# Potential of Fecal Carcinoembryonic Antigen for Noninvasive Detection of Colorectal Cancer: A Systematic Review

**DOI:** 10.3390/cancers15235656

**Published:** 2023-11-30

**Authors:** Xianzhe Li, Lara Stassen, Petra Schrotz-King, Zitong Zhao, Rafael Cardoso, Janhavi R. Raut, Megha Bhardwaj, Hermann Brenner

**Affiliations:** 1Division of Clinical Epidemiology and Aging Research, German Cancer Research Center (DKFZ), 69120 Heidelberg, Germany; xianzhe.li@dkfz-heidelberg.de (X.L.); zitong.zhao@dkfz-heidelberg.de (Z.Z.); rafael.cardoso@dkfz-heidelberg.de (R.C.); megha.bhardwaj@dkfz-heidelberg.de (M.B.); 2Medical Faculty Heidelberg, Heidelberg University, 69120 Heidelberg, Germany; 3National Center for Tumor Diseases (NCT), NCT Heidelberg, a Partnership between DKFZ and University Hospital Heidelberg, 69120 Heidelberg, Germany; lara.stassen@student.maastrichtuniversity.nl (L.S.); petra.schrotz-king@nct-heidelberg.de (P.S.-K.); janhavi.raut@nct-heidelberg.de (J.R.R.); 4Division of Preventive Oncology, German Cancer Research Center (DKFZ) Heidelberg, 69120 Heidelberg, Germany; 5German Cancer Consortium (DKTK), German Cancer Research Center (DKFZ), 69120 Heidelberg, Germany

**Keywords:** carcinoembryonic antigen, colorectal cancer, diagnosis, stool

## Abstract

**Simple Summary:**

Carcinoembryonic antigen (CEA) in serum is widely used as a tumor marker in colorectal cancer (CRC). The levels of fecal CEA (FCEA) are higher than serum CEA (SCEA), especially in the early stages of CRC. In this systematic review, we aimed to provide a comprehensive overview of studies that evaluated FCEA as a biomarker for the noninvasive diagnosis and diagnosis of CRC. All of the few identified studies found statistically significant differences in FCEA levels between the CRC and control groups. Moreover, the diagnostic performance of FCEA surpassed that of SCEA, suggesting a potential role as a novel, easily measurable biomarker for the diagnosis of CRC. However, evidence is still limited to a few, mostly small, studies from clinical settings, and comprehensive evaluation in screening settings is warranted.

**Abstract:**

Carcinoembryonic antigen (CEA) is more abundant in feces than in serum; however, evidence for the role of fecal CEA (FCEA) in the detection of colorectal cancer (CRC) is limited. We conducted a systematic review of studies that evaluated FCEA for the noninvasive detection and diagnosis of CRC. PubMed and Web of Science were searched for relevant studies published until 18 January 2023. Information on publication year, study design, country, study population characteristics, FCEA and serum CEA (SCEA) concentrations, and diagnostic performance was summarized. Two authors independently extracted data and assessed the risk of bias and applicability of each included study. Seven studies published between 1979 and 2021, all conducted in clinical settings and together involving 399 CRC patients and 889 controls, were identified. Significant differences in FCEA concentrations were observed between CRC and control groups in all studies. Methods for detecting FCEA varied, with the electronic chemiluminescence immunoassay (ECLIA) being used in the most recent studies. Reported sensitivities, specificities, and area under the curves of FCEA ranged from 50.0% to 85.7%, 73.0% to 100.0%, and 0.704 to 0.831, respectively. In direct comparisons, the diagnostic performance of FCEA was better than that of SCEA. The potential role of FCEA as a novel, noninvasive, easily measurable biomarker for the diagnosis of CRC requires further evaluation in screening settings.

## 1. Introduction

Colorectal cancer (CRC) is the third most common malignancy and the second most common cause of cancer death, accounting for over 1.9 million new cases and over 0.9 million deaths worldwide in 2020 [1]. Despite significant improvements in treatment, the prognosis of patients with advanced TNM-stage CRC remains poor [2,3]. The 5-year relative survival rate of patients with stage IV CRC in the US was estimated to be at 12%, in contrast to 91% in patients with stage I CRC [4]. Therefore, screening for early-stage CRC is a critical strategy for reducing CRC-associated mortality.

Currently, there are various approaches to CRC screening, but colonoscopy, flexible sigmoidoscopy, and fecal occult blood testing (FOBT, i.e., fecal immunochemical test (FIT)) are the most widely used [5,6]. Although colonoscopy is the gold standard for the diagnosis of CRC and its precursors, the compliance rate for colonoscopy is low due to its invasiveness, high cost, and requirement for extensive bowel preparation [7,8,9,10]. Compared with colonoscopy, flexible sigmoidoscopy is less invasive, less costly, and does not require complete bowel preparation and sedation; however, its major drawback is the inability to visualize neoplasms in the proximal colon [11,12]. FITs are less costly and more convenient than colonoscopy but have relatively low sensitivity for detecting advanced adenoma and early-stage CRC [13,14]. The sensitivity of FIT for CRC screening ranges from 68.8% to 81.3%, whereas its sensitivity for detecting advanced adenomas is much lower (18.0–43.5%) [15]. A large screening study with more than 20,000 participants showed poor performance of FIT for detecting advanced adenoma, with a sensitivity below 22% [16]. Therefore, a novel, cost-effective, noninvasive, and easily measurable CRC screening test, with high sensitivity and specificity, would be highly desirable.

Carcinoembryonic antigen (CEA) is widely used as a tumor marker in most gastrointestinal cancers. It is formed in the cytoplasm and can be detected in serum, cerebrospinal fluid, urine, and feces [17]. Serum CEA (SCEA) levels are low in healthy individuals and may rise 4 to 8 months before cancer-related symptoms develop [18]. However, SCEA is not widely used in CRC screening because several studies have found that it does not have sufficient sensitivity and specificity to diagnose CRC, and it is more commonly used to monitor tumor recurrence [17,19].

Feces composed of undigested food, endogenous secretions, microbiota, and exfoliated host cellular components can be used to assess the entire intestinal environment, including the occurrence of CRC and its biological effects on epithelial cells [20]. As stool is a rich source of cells derived from the gastrointestinal tract, several oncoproteins derived from intact tumor cells or tumor cell debris are present in the stool of patients with CRC. The concentrations of fecal CEA (FCEA) are higher than those of SCEA, especially in the early stages of CRC, prompting researchers to advocate using FCEA for the diagnosis of CRC.

Several studies suggested FCEA to be more sensitive than SCEA for early CRC detection [21,22,23]. However, no systematic review has evaluated the potential of FCEA in the detection of CRC. In this systematic review, we aimed to provide a comprehensive overview of studies that evaluated FCEA as a biomarker for the noninvasive detection of CRC.

## 2. Materials and Methods

This systematic review was conducted in accordance with the Preferred Reporting Items for Systematic Reviews and Meta-Analyses protocols (PRISMA-P) [24]. The protocol has not been registered.

### 2.1. Data Sources and Search Strategy

The PubMed and Web of Science databases were searched for relevant articles published up to 18 January 2023. The search terms included (CEA OR carcinoembryonic antigen OR carcinoembryonic AND antigen) AND (faeces OR feces OR stool OR stooling OR stools) AND (CRC OR colorectal cancer OR colorectal AND cancer OR colorectal neoplasms OR colorectal AND neoplasms). Duplicate hits were removed.

### 2.2. Study Selection

Studies were eligible for inclusion in this systematic review if they met the following inclusion criteria: examination of CEA in stool samples from patients with CRC at various stages compared with control groups of individuals without CRC. The search was restricted to human studies published in English. The first step in the selection of eligible studies was based on reading the title and abstract. Articles were excluded if they were (1) not based on stool samples, (2) not full papers, (3) not related to the topic, (4) not original articles (e.g., case reports) or reviews, or (5) not in English. Full texts of the remaining articles were reviewed and included if deemed relevant. Finally, studies that did not report the key study characteristics, stratified results, or sufficient data for calculation were excluded.

### 2.3. Data Extraction

Two authors (X.L. and L.S.) independently extracted data from eligible studies. Descriptive key characteristics of eligible studies, including publication year, study design, country, study population characteristics, FCEA, and SCEA concentrations, were extracted. In addition, we extracted descriptions of the stool sampling and quantitative FCEA detection methods. Furthermore, sensitivity, specificity, and area under the curve (AUC) for evaluating the diagnostic performance of FCEA and SCEA were extracted.

### 2.4. Risk of Bias and Applicability Assessment of Each Study

The Quality Assessment of Diagnostic Accuracy Studies 2 (QUADAS-2) [25] tool was used to assess the quality of each included study in terms of patient selection, index test, reference standard, flow, and timing. In QUADAS-2, each domain is evaluated for risk of bias, and the first three domains are evaluated for applicability. The risk of bias and applicability assessment for each study was rated as “high risk”, “low risk”, or “unclear risk”. Two authors (X.L. and L.S.) independently performed the assessments using Review Manager 5.4.1 (The Cochrane Collaboration, London, UK, 2020).

## 3. Results

### 3.1. Literature Search Results

The literature search and selection process are shown in Figure 1, including four stages of “identification”, “screening”, “eligibility”, and “included”. After removing duplicates, 155 articles were identified. On inspection of titles and abstracts, 136 articles were excluded because they either did not evaluate fecal samples, were not full papers, were not related to the topic, were non-original, or were not in English. We selected 19 articles for full-text assessment. Of these, 12 articles were excluded because they did not provide relevant data on key study characteristics such as FCEA values. Finally, seven studies on CEA levels in feces published up to 18 January 2023 were included in this systematic review.

### 3.2. Study Characteristics

Table 1 summarizes the key characteristics of the studies included in this review. The studies consisted of seven case–control studies [21,22,23,26,27,28,29], involving a total of 399 CRC cases and 889 controls, including patients with gastric cancer (GC), non-gastrointestinal cancer (NGIC), adenomatous polyposis coli (APC), benign gastrointestinal disorders (BGID), and healthy controls (HCs). The NGIC group included patients with head and neck, hepatobiliary, pancreatic, breast, lung, reproductive system, esophageal, and kidney cancers. The year of publication ranged from 1979 to 2021, with four studies reported between 1979 and 1989, one study in 2003, and two studies in 2021. Six studies were conducted in Asian countries (China, Japan, and South Korea) and one in a European country (the United Kingdom). Sample sizes were rather small in the earlier studies, and the largest two were in the most recent studies from China, which included 436 [21] and 298 [22] participants, with potentially overlapping study populations. Only three studies provided data on age and sex [21,22,29].

### 3.3. FCEA and SCEA Concentrations in Different Groups

As shown in Table 1, statistically significant differences in FCEA concentrations were observed between the HC and CRC groups in all studies (*p* < 0.05). Two studies found that compared with the CRC group, the FCEA concentration of the NGIC group was significantly different (*p* < 0.001), but there was no statistically significant difference between the CRC and APC groups [21,22]. Three studies found that compared with the CRC group, the BGID group also had statistically significant differences in FCEA (*p* < 0.01) [23,26,27]. Kim et al. [23] found that, compared with the CRC group, there was no statistically significant difference in FCEA in either the advanced GC (*p* = 0.879) or the early GC group (*p* = 0.909). In addition, only three studies reported on SCEA concentrations. Two studies found that compared with the CRC group, there were statistically significant differences in SCEA concentrations in the NGIC, APC, and HC groups [21,22]. Another study found that compared with the CRC group, except for advanced GC, mean SCEA concentrations in the early GC, BGID, and HC groups were significantly different [23]. Although the units of measurement differed, the CEA concentrations in feces were much higher than those in blood.

### 3.4. Different Methods for Stool Sampling and Fecal CEA Quantitative Detection

All the included studies provided descriptions of stool sampling and FCEA quantitative detection methods, as summarized in Table 2. The two studies reported by Li et al. [22] and Li et al. [21] in 2021 were from the same research group in China; they used similar approaches in terms of stool sampling and FCEA quantitative detection. The studies conducted by Kitsukawa et al. [28] and Fujimoto et al. [29] also used the same methods. However, in general, there were notable variations in the detailed methods of stool sampling and FCEA detection across the studies. Furthermore, it is noteworthy that over time, the quantitative detection method for FCEA has evolved from radioimmunoassay [28,29] to enzyme-linked immunoassay (ELISA) [27] and, subsequently, to electronic chemiluminescence immunoassay (ECLIA) [21,22], which is currently one of the most widely used methods for protein biomarker detection.

### 3.5. Diagnostic Performance of Fecal CEA Compared with Serum CEA

Among all the included studies, five assessed or calculated the diagnostic performance (sensitivity, specificity, and AUC) of FCEA or SCEA [21,22,23,26,28]. The results are summarized in Table 3. The sensitivity of FCEA ranged from 50% to 85.7% in four studies, whereas the sensitivity of SCEA ranged from 29.8% to 39.3% in three studies [21,23,28]. The specificity of FCEA ranged from 73.0% to 100% in three studies [21,22,23,26,28], whereas the specificity of SCEA ranged from 90.0% to 98.0% in two studies [21,23]. Additionally, AUC values were reported in two studies [21,22], ranging from 0.704 to 0.831 for FCEA, and from 0.525 [23] to 0.861 for SCEA.

### 3.6. Assessment of Risk of Bias and Applicability across Studies

The results of the risk of bias and applicability assessment, based on the QUADAS-2 criteria, are summarized in Figure 2. The primary source of potential bias and applicability concerns in the studies included in the review was the selection of study participants. All studies were case–control studies, and recruited participants had already been diagnosed in clinical settings rather than in prospective true screening settings, so the “Patient Selection” for all included studies was considered high risk in risk of bias and applicability. However, the “Reference Standard” was low risk for both bias and applicability for all included studies, except one reported by Sugano et al. [26]. In four of the seven studies, the “Flow and Timing” had an unclear risk because it was unclear whether there was an appropriate interval between the measurement of FCEA and colonoscopy/pathology.

## 4. Discussion

CRC often grows slowly through a multistep process, involving a series of histological, morphological, and genetic changes that accumulate over time. This allows for the screening and detection of early-stage precancerous lesions before they become cancerous in individuals at average risk of CRC [30]. Serum CEA has long been one of the most widely used and evaluated biomarkers for detecting CRC. Although it is often elevated, especially in the advanced stages of the disease, low sensitivity for detecting early stages of the disease limited its use for CRC diagnosis and screening [17,19]. However, CEA can be detected not only in serum but also in feces [31]. Feces are a rich source of cells derived from the gastrointestinal tract, which can be used to measure proteins such as CEA originating in the intestinal mucosa and assess the occurrence of CRC [32]. Kim et al. [23] speculated that if adequate numbers of cancer cells or their products are mixed in the feces, the amount of fecal CEA should be higher in patients with CRC than in normal controls.

However, surprisingly few studies have reported on the potential use of FCEA for CRC detection. The few studies, all of which were conducted in clinical settings, consistently found statistically significant differences in FCEA levels between the CRC and HC groups. In addition, FCEA levels in the CRC group were higher than those in the other disease groups, including APC, NGIC, and BGID. Studies reporting both FCEA and SCEA also found that the levels of CEA in feces were higher than those in the serum. The reason may be that CRC tumor cell-derived CEA is transported from the portal vein to the liver and then decomposed, decreasing CEA concentration in the blood, but it is more likely to be enriched in feces without significant degradation [22]. Li et al. [22] further found that the concentration of FCEA was correlated with tumor size: the concentration of FCEA in patients with CRC with a tumor diameter ≥5 cm was significantly higher than that in those with a tumor diameter <5 cm. A possible reason for this is that the larger the tumor size, the more CRC tumor cells secrete CEA, which is then released into the gut and enriched in feces.

Among the included studies, five evaluated the diagnostic performance of both FCEA and SCEA, and the diagnostic performance (sensitivity, specificity, and AUC) of FCEA was better than that of SCEA, suggesting that FCEA might be a better biomarker than SCEA for the diagnosis of CRC. The FIT for hemoglobin is currently the best-established test for the noninvasive detection of CRC. However, only one study compared the diagnostic performance of FCEA with that of an FIT. Li et al. [21] showed that the AUC value of FCEA for CRC diagnosis was 0.802, which was lower than 0.903 using the FIT. In an older study from 2003, Kim et al. [23] found that the sensitivity for CRC diagnosis of FCEA was 60.0%, and the specificity was 98.2%, both of which were higher than 37.5% and 91.3% using a chemical fecal occult blood test available at that time. The different patterns may primarily be explained by the different types of fecal occult blood tests. Further research should explore if FCEA, by itself or in combination with FIT or other promising fecal tests, might have a potential role in stool-based CRC screening. Compared with colonoscopy, stool-based CRC screening tests may have the advantages of being noninvasive and easily incorporated into routine clinical practice [33]. They do not require bowel preparation or sedation, making them a convenient option for patients.

In this review, we also summarized the methods used for stool sampling and quantitative detection of FCEA in various studies. Except for the studies from the same research group, differences were observed in the specific methodologies used for stool sampling and FCEA detection among the studies. In addition, it is important to highlight the evolution of quantitative methods for FCEA. Initially, a radioimmunoassay was mainly used [28,29], which was later replaced by ELISA [27]. Then, the two most recent studies from the same group in China both used ECLIA [21,22]. Compared to radioimmunoassay and ELISA, ECLIA may offer higher sensitivity and accuracy in detection, as well as simpler operational procedures. With the development of technology, quantitative detection methods for fecal CEA will continue to evolve. Therefore, more accurate and convenient detection methods may be developed.

Generally, when identifying diagnostic biomarkers for CRC screening, study participants need to be representative of the target population and adhere to the recruitment criteria [34]. Differences in the characteristics of the study populations may result in discrepancies in the diagnostic performances of the reported biomarkers. Additionally, the most suitable setting for identifying diagnostic biomarkers is real screening settings. However, all studies included in this review were case–control studies, and the participants had already been diagnosed in clinical settings, not in prospective true screening settings, which may introduce potential spectrum bias [35]. The lack of further external validation of the results of these studies in different settings may also have contributed to potential overestimation and overoptimism [36]. Among the included studies, six originated from Asian countries (China, Japan, and South Korea) and one was from a European country (the United Kingdom), which could potentially restrict the generalizability of the findings to other populations. None of the included studies provided the number of patients who failed to submit stool samples or the number of patients who submitted unusable samples. Therefore, we were unable to assess the failure rates of the FCEA or SCEA test. Moreover, only two studies [27,29] were not typical diagnostic studies because the diagnostic performance results (sensitivity, specificity, and AUC) were not provided. However, to ensure the consistency of the evaluation, we still chose the QUADAS-2 criteria to evaluate the two studies. In addition, the value of fecal CEA may be questionable for the diagnosis of CRC precancerous lesions in clinical practice due to the lack of sufficient evidence of its effectiveness. Further large-scale screening cohorts are needed to settle this issue.

Although this systematic review provided a comprehensive overview of studies investigating and assessing FCEA as a potential biomarker for CRC screening, it has some limitations. First, despite the exhaustive search conducted by two independent reviewers in two reputable databases and meticulous cross-referencing, it remains possible that relevant studies, particularly those in languages other than English, might have been inadvertently overlooked, thus introducing language bias. Second, due to the significant heterogeneity observed among the studies, it was not feasible to conduct meta-analyses to combine the study results. Finally, there was no single original article published about the detection or diagnosis of CRC via FCEA between 2004 and 2020. There might be some publication bias here (i.e., studies might have been conducted but they were simply unpublished due to the absence of significant results).

## 5. Conclusions

To the best of our knowledge, this systematic review is the first to provide a comprehensive overview of studies that assessed FCEA as a potential biomarker for the noninvasive detection of CRC. All of the few studies reported so far found statistically significant differences in FCEA levels between the CRC and control groups. Moreover, the diagnostic performance of FCEA surpassed that of SCEA, suggesting that it could be a potentially useful, noninvasive, and easily measurable biomarker for the diagnosis of CRC. Nevertheless, evidence is still very limited and essentially restricted to a few, mostly small, case–control studies from clinical settings, with only three studies (including two recent studies from the same group) published since 1990. Further research, in particular large-scale prospective studies with comprehensive analyses of diagnostic performance of FCEA compared to or in combination with other tests, such as FIT, is needed to explore a potential role for CRC diagnosis and screening. Such studies should not only address the detection of CRC but also precursors of CRC, whose detection is essential for CRC prevention. Further research should also address the potential impact of using FCEA or SCEA, alone or in combination with FIT, on the long-term outcomes of screening, such as CRC incidence and mortality.

## Figures and Tables

**Figure 1 cancers-15-05656-f001:**
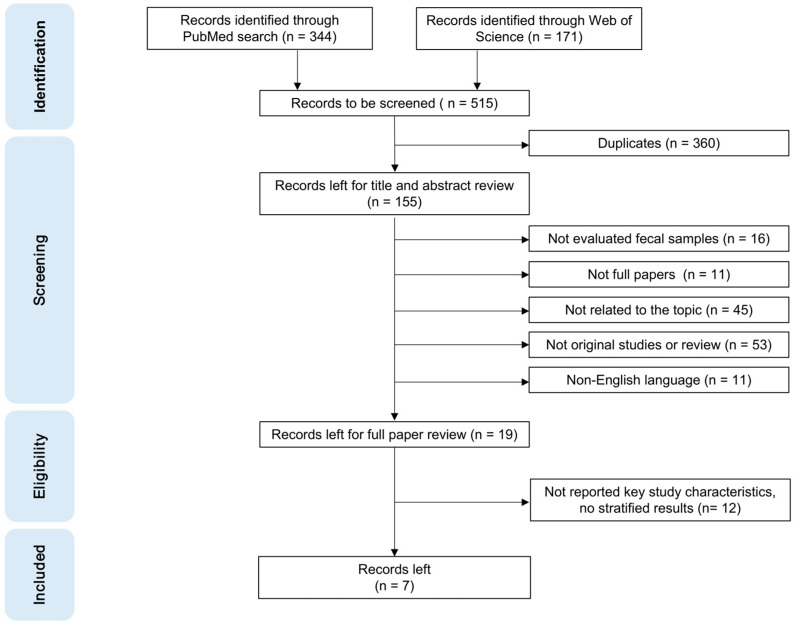
PRISMA flow diagram for the literature search process for records identified using PubMed and Web of Science databases.

**Figure 2 cancers-15-05656-f002:**
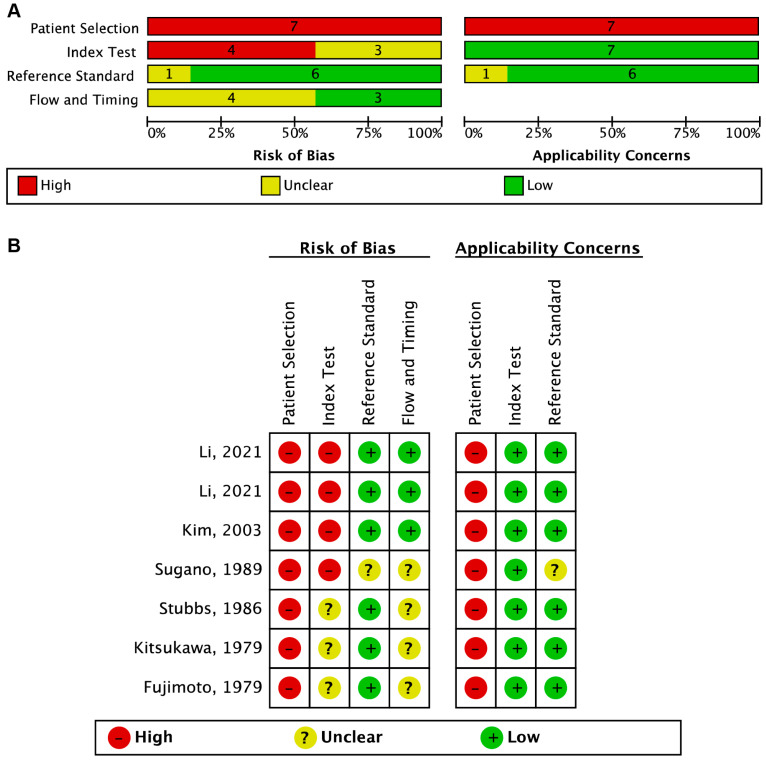
The risk of bias and applicability assessment based on the QUADAS-2 criteria. (**A**) Risk of bias and applicability concerns graph: review authorsʹ judgments about each domain presented as percentages of the included studies. (**B**) Risk of bias and applicability concerns summary: review authors’ judgments about each domain for each included study [21,22,23,26,27,28,29].

**Table 1 cancers-15-05656-t001:** Key characteristics of all included studies.

First Author, Year	Study Type	Country	Race	Study Groups	Population (n)	Mean Age (Range)	Sex, Female (%)	Fecal CEA (ng/mg)	*p* Value	Serum CEA (ng/mL)	*p* Value
Li, 2021 [21]	Case-control	China	Asian	CRC	166	59 (25–87)	70 (42.2)	148.4 (80.91–245.7)	ref	3.28 (1.95–8.73)	ref
				APC	46	53 (21–73)	18 (39.1)	115.7 (65.14–186.6)	0.096 ^a^	1.75 (1.27–2.79)	<0.001 ^a^
				NGIC	60	52 (21–82)	24 (40.0)	97.58 (56.83–130.6)	<0.001 ^a^	2.29 (1.22–4.44)	0.002 ^a^
				HC	164	40 (20–65)	91 (55.5)	53.54 (31.48–86.51)	<0.001 ^a^	1.47 (0.95–2.25)	<0.001 ^a^
Li, 2021 [22]	Case-control	China	Asian	CRC	115	59 ± 11	51 (44.4)	149.76 (81.0–240.9)	ref	3.28 (1.75–9.92)	ref
				APC	35	46 ± 12 ^b^	89 (48.6) ^b^	113.58 (63.9–182.97)	0.167	1.97 (1.34–3.21)	<0.001
				NGIC	46			83.58 (53.42–135.29)	<0.001	1.84 (1.17–4.67)	0.033
				HC	102			46.19 (26.17–84.72)	<0.001	1.50 (1.09–2.17)	<0.001
Kim, 2003 [23]	Case-control	South Korea	Asian	CRC	28			45.2 ± 63.8	ref	8.87 ± 13.28	ref
				Invasive GC	19			42.9 ± 38.8	0.879	5.09 ± 8.57	0.242
				Early GC	13			42.0 ± 89.6	0.909	3.13 ± 2.51	0.035
				BGID	20			4.5 ± 8.2	0.002	1.66 ± 2.03	0.008
				HC	240			3.7 ± 3.5	0.002	1.14 ± 1.0	0.005
Sugano, 1989 [26]	Case-control	Japan	Asian	CRC	24			0.46 ± 0.51 μg/mL	ref	/	/
				BGID	55			0.17± 0.19 μg/mL	<0.001	/	/
				HC	15			0.04 ± 0.03 μg/mL	<0.001	/	/
Stubbs, 1986 [27]	Case-control	United Kingdom	European	CRC	24			10.43 ± 2.39	ref	/	/
				BGID	14			7.12 ± 1.39	<0.001	/	/
				HC	20			5.43 ± 1.95	<0.001	/	/
Kitsukawa, 1979 [28]	Case-control	Japan	Asian	CRC	20			215 ± 93 ng/g	ref	/	/
				HC	20			77.7 ± 42.9 ng/g	<0.001	/	/
Fujimoto, 1979 [29]	Case-control	Japan	Asian	Dukes A	4	55 (41–74) ^c^	12 (54.5) ^c^	193 ± 51 ng/g	0.014	/	/
				Dukes B	3			178 ± 73 ng/g	0.135	/	/
				Dukes C	9			213 ± 90 ng/g	0.002	/	/
				Liver metastasis	6			267 ± 95 ng/g	0.004	/	/
				HC	20	51 (40–67)	2 (10.0)	78 ± 42 ng/g	ref	/	/

Abbreviations: APC, adenomatous polyposis coli; BGID, benign gastrointestinal disorders; CRC, colorectal cancer; GC, gastric cancer; HC, healthy controls; NGIC, non-gastrointestinal cancer; n, number; ref: reference. Annotation: ^a^ The value is calculated based on the data in the article; ^b^ The data represents the two groups of NGIC and HC; ^c^ The data represents the four groups of Dukes A, Dukes B, Dukes C, and Liver metastases. Data is shown as mean ± SD, or median and quartiles.

**Table 2 cancers-15-05656-t002:** Summary of different methods for stool sampling and fecal CEA quantitative detection.

Study	Stool Sampling and Processing Methods	Fecal CEA Detection Methods
Li, 2021 [21]	0.1 mg of fresh feces was collected by fecal collection tubes from three different locations, homogenized for 2 min and then centrifuged for 10 min at 10,000 r/min. The supernatant was retained. The samples were filtered, if necessary.	ECLIA Kit (Roche Diagnostics, Mannheim,, Germany)
Li, 2021 [22]	Stool samples were collected from CRC patients within 7 days before surgery or treatment, and from healthy individuals on the day of the physical examination. Loose stools, watery stools, blood stools, or hard stools are excluded. Other steps including the amount of stool were the same as Li et al.’s study above.	ECLIA (Roche Diagnostics, Mannheim, Germany)
Kim, 2003 [23]	Stool samples were collected within 1 week after the histologic diagnoses were made. About 80 mg of each stool sample were added to 800 µL of pH 7.4 PBS. After twice freezing and thawing procedures, the mixtures were filtered through a polyvinyl alcohol sponge filter.	Automated immunoassay system (Elecsys 2010, Roche Diagnostics, Mannheim, Germany)
Sugano, 1989 [26]	Stool samples (20 mg) were dissolved in 5 mL of 0.1 M, pH 7.4 PBS containing 0.15 M NaCl and 0.2% sodium azide. After centrifugation at 3000 rpm for 15 min, supernatants were recovered.	A “forward sandwich” radioimmunometric assay
Stubbs, 1986 [27]	One entire stool sample was collected and weighed before surgery, homogenized with 100 mL 0–9% NaCl and an aliquot weighing about 500 mg stool was taken. 5 mL PBS, pH 7.1, were added, mixed thoroughly and incubated at 80 °C for 10 min in a water bath. After centrifuging for 15 min at 600× *g*, the supernatant was added to a further 5 mL pH 7.1 PBS and recentrifuged for 10 min at 600× *g*. Then the supernatant was filtered through a number 1 Whatman filter paper.	ELISA
Kitsukawa, 1979 [28]	Stool samples were collected without barium, blood and diarrhea. About 500 mg stools were put into ten-fold the quantity of 0.1 M acetate buffer (pH 5.0) and were mixed by stirring. Then the stools were centrifuged at 2500× *g* for 10 min. The supernatant was diluted with an equal volume of acetate buffer, incubated at 85 °C for 10 min in a water bath and then centrifuged for 5 min at 2500× *g.*	Radioimmunoassay utilizing the “one step sandwich method”
Fujimoto, 1979 [29]	The same as Kitsukawa et al.’s study above.	Radioimmunoassay utilizing the “one step sandwich method”

Abbreviations: CRC, colorectal cancer; ECLIA, electronic chemiluminescence immunoassay; ELISA, enzyme-linked immunoassay; PBS, phosphate buffer saline.

**Table 3 cancers-15-05656-t003:** Summary of diagnostic performance of fecal CEA compared with serum CEA.

Study	Groups	Fecal CEA			Serum CEA		
		SEN (%)	SPE (%)	AUC	SEN (%)	SPE (%)	AUC
Li, 2021 [21]	CRC vs. NGIH+HC	/	/	0.802	/	/	0.757
	APC vs. NGIH+HC	/	/	0.704	/	/	0.525
	CRC+APC vs. NGIC+HC	/	/	0.781	/	/	0.861
	Stage I CRC+ APC vs. NGIC+HC	/	/	0.729	/	/	0.589
Li, 2021 [22]	CRC vs. NGIC+HC	76.50	73.00	0.802	38.30	91.00	0.735
	Stage I+II CRC vs. HC	78.70	73.50	0.831	29.80	98.04	0.750
Kim, 2003 [23]	CRC vs. HC	85.70	92.92	/	39.29	96.66	/
	CRC vs. BGID	85.70	95.00	/	39.29	90.00	/
Sugano, 1989 [26]	CRC vs. HC	50.00 ^a^	100.00 ^a^				
Kitsukawa, 1979 [28]	CRC vs. HC	85.00 ^a^	/	/	35.00 ^a^	/	/

Abbreviations: APC, adenomatous polyposis coli; AUC, area under the curve; BGID, benign gastrointestinal disorders; CRC, colorectal cancer; HC, healthy controls; NGIC, non-gastrointestinal cancer; n, number; SEN, sensitivity; SPE, specificity. Annotation: ^a^ The value is calculated based on the data in the article.

## Data Availability

The data presented in this study are available in article.

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
