# Peer review of "Potential of Fecal Carcinoembryonic Antigen for Noninvasive Detection of Colorectal Cancer: A Systematic Review"

_cancers, 2023, doi:10.3390/cancers15235656_

Round 1
Reviewer 1 Report
Comments and Suggestions for Authors
This is an interesting review. The writing is clear and straightforward. The author presents the material in an organized fashion and follows the general guidance of systematic review. However, several parts of the manuscript need more detail and clarity.
This study aims to provide a comprehensive overview of studies that evaluated FCEA as a biomarker for the noninvasive early detection and diagnosis of CRC. There are 7 studies selected in this review. Two of them presented data comparing FCEA with FOBT. Lack of FOBT data, the data in the table of this paper only included those of FCEA and SCEA from selected papers. To evaluate biomarkers for the noninvasive early detection of CRC, it would be appropriate to compare, the FCEA test, a focal-based test, in this article, with a currently best-established focal test, FOBT, even though the evidence now is limited and high-quality original research and better systematic review of this issue could be achieved in the future. In the discussion part, I would also suggest stating the result of the comparison of FCEA with FOBT in those two papers and discussing the potential implications.
The author should include more basic characteristics of the selected study population, eg. Age, Sex, etc.
To evaluate the outcome of selected studies, other than accuracy outcomes (sensitivity, specificity), it would be favorable to include failure rates of the FCEA test, SCEA, and FOBT, for example, the number of patients who failed to submit stool samples, the number of patients who submitted unusable samples.
It is always important to indicate the clinical value of the study, so, in the discussion part, it should be addressed whether the author identified studies on the long-term effects of testing FCEA on mortality, morbidity, and health-related quality of life. If not, he should further suggest prospective randomized controlled trials would be followed in the future.
In the discussion, the author pointed out that the detection method for FCEA evolves over a while and therefore more accurate and convenient detection methods would emerge. I would suggest briefly mentioning how the newer method (ECLIA) superior to older ones (radioimmunoassay, ELISA).
Search terms could be added, eg, “fecal OR fecal” to enlarge the literature search results. There is a typo in search terms—(feces OR feces OR stool OR stooling OR stools) where two “feces” are used. Table 1, should be addressed which statistic is used to describe fecal CEA and serum CEA. Is it a mean value of CEA?
Author Response
Thank you very much for your constructive comments. We have revised this systematic review explicitly based on your comments. Please see the following responses and revised manuscript for details.
- This study aims to provide a comprehensive overview of studies that evaluated FCEA as a biomarker for the noninvasive early detection and diagnosis of CRC. There are 7 studies selected in this review. Two of them presented data comparing FCEA with FOBT. Lack of FOBT data, the data in the table of this paper only included those of FCEA and SCEA from selected papers. To evaluate biomarkers for the noninvasive early detection of CRC, it would be appropriate to compare, the FCEA test, a focal-based test, in this article, with a currently best-established focal test, FOBT, even though the evidence now is limited and high-quality original research and better systematic review of this issue could be achieved in the future. In the discussion part, I would also suggest stating the result of the comparison of FCEA with FOBT in those two papers and discussing the potential implications.
A: Thank you very much for your constructive comment. We agree with you. Actually, when we conducted this systematic review, we also wanted to compare the diagnostic performance of FCEA with FOBT or FIT for CRC. However, as you can see, there are very few relevant published studies, and only two of the seven included studies showed the diagnostic performance data of FOBT or FIT, so we did not show the data of them in Table 3. However, based on your suggestion, we now address the data from the two studies comparing the diagnostic performance of FCEA with that of FOBT or FIT for CRC in detail in the discussion. (See revised manuscript page 10, lines 248-254).
- The author should include more basic characteristics of the selected study population, eg. Age, Sex, etc.
A: Thank you very much for the comment. Based on your suggestion, we have added Age and Sex data to Table 1 (See revised manuscript page 4, lines 148-149 and page 5).
- To evaluate the outcome of selected studies, other than accuracy outcomes (sensitivity, specificity), it would be favorable to include failure rates of the FCEA test, SCEA, and FOBT, for example, the number of patients who failed to submit stool samples, the number of patients who submitted unusable samples.
A: Thank you very much for the comment. Unfortunately, none of the included studies provided the number of patients who failed to submit stool samples, or the number of patients who submitted unusable samples. Therefore, we were unable to assess the failure rates of the FCEA test, SCEA, and FOBT. We now address this important point in the discussion (See revised manuscript page 11, lines 283-285).
- It is always important to indicate the clinical value of the study, so, in the discussion part, it should be addressed whether the author identified studies on the long-term effects of testing FCEA on mortality, morbidity, and health-related quality of life. If not, he should further suggest prospective randomized controlled trials would be followed in the future.
A: Thank you very much for the comment. None of the included studies provided information on the long-term effects of testing FCEA on mortality, morbidity, and health-related quality of life. We now emphasize the importance of future studies on these endpoints in the final conclusions. (See revised manuscript page 11, lines 315-319).
- In the discussion, the author pointed out that the detection method for FCEA evolves over a while and therefore more accurate and convenient detection methods would emerge. I would suggest briefly mentioning how the newer method (ECLIA) superior to older ones (radioimmunoassay, ELISA).
A: Thank you very much for the comment. We now briefly describe how the newer method (ECLIA) is superior to older ones (radioimmunoassay, ELISA) in the discussion. Compared to radioimmunoassay and ELISA, ECLIA may offer higher sensitivity and accuracy in detection, and simpler operational procedures. (See revised manuscript page 10, lines 266-268).
- Search terms could be added, eg, “fecal OR fecal” to enlarge the literature search results. There is a typo in search terms—(feces OR feces OR stool OR stooling OR stools) where two “feces” are used. Table 1, should be addressed which statistic is used to describe fecal CEA and serum CEA. Is it a mean value of CEA?
A: We tried inclusion of “fecal OR fecal” as additional search terms which did not lead to identification of further studies. Thank you very much for alerting us of the typo which we have corrected (See revised manuscript page 2, line 92). In addition, we have added information on the statistics (mean ± SD, median and quartiles) describing fecal CEA and serum CEA in the legend below Table 1. (See revised manuscript page 6, lines 154-155).
Reviewer 2 Report
Comments and Suggestions for Authors
In this article, the authors conducted a systematic review of fecal carcinoembryonic antigen (CEA) in the detection of early colorectal cancer (CRC). This topic has some value in developing new tools in colorectal cancer screening. However, I have several concerns about this paper:
1. Two of the included studies (from the same group, Li et al. 2021) may have a overlapping cohort since the case enrollment period in these two original studies were 2019-2020 and 2019, respectively. This problem have to be clearly addressed to avoid serious bias.
2. The fecal CEA level appear to vary from studies even in healthy control and colorectal cancer partients and hence a reference value is difficult to be determined. In addition, few studies provided comparison between fecal and serum CEA. These factors may preclude further clinical use until any further large cohort study settle this issue. Hence, the conclusion need to be varified in further studies.
3. The merit of fecal occult blood, fecal DNA and colonosopy not only includes early cancer detection but also be helpful in detecting precancer lesions, which can be beneficial in prevention of colorectal cancer incidence and mortality. On the otherhand, fecal CEA has not proved its ability to detect precancer lesions. Thus, the value of fecal CEA may be questionable in the clinical practice.
4. There was only one out of seven article provided the fecal DNA level in stage I CRC. Therefore, it may not be sufficient to conclude fecal CEA is help to "early detection" of CRC.
Comments on the Quality of English LanguageThe English writting was clear and fluent.
Author Response
Thank you very much for your constructive comments. We have revised this systematic review explicitly based on your comments. Please see the following responses and revised manuscript for details.
- Two of the included studies (from the same group, Li et al. 2021) may have a overlapping cohort since the case enrollment period in these two original studies were 2019-2020 and 2019, respectively. This problem have to be clearly addressed to avoid serious bias.
A: Thank you very much for your constructive comment. Actually, we had previously discussed within our research group about the including the two studies by Li et al. The concern addressed by the reviewer would be highly relevant as it could lead to serious bias if meta-analyses were performed. However, this was not done in our paper. Furthermore, although the samples used by these two studies seem to overlap, the purposes of the two studies are different. The first study [21] with more samples aimed to identify the diagnostic potential of FCEA for the detection of early lesions of CRC, including APC and asymptomatic CRC. The second study [22] was mainly to compare the diagnostic potential of FCEA and SCEA for diagnosis of CRC. Based on the above reasons, we finally decided to include both studies in this systematic review. Nevertheless, we now explicitly address the potential overlap of the study populations (See revised manuscript page 4, lines 146-148).
- The fecal CEA level appear to vary from studies even in healthy control and colorectal cancer partients and hence a reference value is difficult to be determined. In addition, few studies provided comparison between fecal and serum CEA. These factors may preclude further clinical use until any further large cohort study settle this issue. Hence, the conclusion need to be varified in further studies.
A: Thank you very much for the comment. We wholeheartedly concur with your statement and now explicitly address this important point in the conclusion (See revised manuscript page 11, lines 309-319).
- The merit of fecal occult blood, fecal DNA and colonosopy not only includes early cancer detection but also be helpful in detecting precancer lesions, which can be beneficial in prevention of colorectal cancer incidence and mortality. On the otherhand, fecal CEA has not proved its ability to detect precancer lesions. Thus, the value of fecal CEA may be questionable in the clinical practice.
A: Thank you very much for the comment. Again, we agree and now explicitly address this important point in the discussion and the conclusion( See revised manuscript page 11, lines 289-291 and lines 315-316).
- There was only one out of seven article provided the fecal DNA(CEA?) level in stage I CRC. Therefore, it may not be sufficient to conclude fecal CEA is help to "early detection" of CRC.
A: Thank you very much for the comment. We agree with your perspective. Based on the seven studies included in this systematic review, it is not sufficient to draw a definitive conclusion that fecal CEA can contribute to "early detection" of CRC. So we have modified "early detection of CRC" to "diagnosis of CRC" at the corresponding part in the revised manuscript.
Round 2
Reviewer 2 Report
Comments and Suggestions for Authors
The authors have revised sufficiently according to previous comments. I have no further questions.